# Mechanism of Precipitate Microstructure Affecting Fatigue Behavior of 7020 Aluminum Alloy

**DOI:** 10.3390/ma13153248

**Published:** 2020-07-22

**Authors:** Zhaojun Shan, Shengdan Liu, Lingying Ye, Yiran Li, Chunhua He, Jin Chen, Jianguo Tang, Yunlai Deng, Xinming Zhang

**Affiliations:** 1School of Materials Science and Engineering, Central South University, Changsha 410083, China; shanzhaojun@csu.edu.cn (Z.S.); lingyingye@csu.edu.cn (L.Y.); 183112154@csu.edu.cn (C.H.); cchenjin@csu.edu.cn (J.C.); jgtang@csu.edu.cn (J.T.); luckdeng@csu.edu.cn (Y.D.); xmzhang@csu.edu.cn (X.Z.); 2Key Laboratory of Nonferrous Metal Materials Science and Engineering, Ministry of Education, Central South University, Changsha 410083, China; 3School of Civil Engineering, Harbin Institute of Technology, Harbin 150001, China; 1190501525@stu.hit.edu.cn

**Keywords:** 7020 aluminum alloy, artificial aging, precipitate, fatigue strength, fatigue crack growth

## Abstract

The effect of different precipitate microstructures obtained by different heat treatments on fatigue behavior of 7020 aluminum alloy was investigated. The fine Guinier Preston I (GPI) zones in the under-aged alloy can be repeatedly sheared by dislocations produced in cyclic loading, making the fatigue crack initiate difficultly and fatigue crack path propagate tortuously. Fatigue strength and fatigue crack propagation resistance of the alloy with shearable precipitates are much higher than those of the alloy with unshearable precipitates. The peak-aged alloy with continuous grain boundary precipitate (GBP) and narrow precipitate free zone (PFZ) is prone to initiate fatigue cracks and reduce fatigue strength. With the growth of unshearable precipitates, the fatigue strength of the alloy firstly increases and then decreases. Precipitates with moderate size in the over-aged alloy improve the roughness-induced crack closure (RICC) effect. Soft matrix with appropriate width between the precipitates can promote the slip reversibility and relax the crack tip stress. The fatigue strength of the moderately over-aged alloy reaches to 122.1 MPa at 10^7^ cycles of loading, and the fatigue crack growth rate (FCGR) is 35.6% slower than that of the peak-aged alloy at Δ*K* of 10 MPa·m^1/2^.

## 1. Introduction

7020 aluminum alloy belongs to Al-Zn-Mg series heat-treatable aluminum alloy. The alloy has been widely used in high-speed trains and new energy vehicles because of its high strength-weight ratio, good forming performance and excellent welding properties [1,2]. Underframe structural materials of high-speed train have been used in high-load, frequent acceleration and deceleration and other alternating stress environments for a long time, which require good fatigue performance to ensure the service life of the train. Precipitate characteristics in Al-Zn-Mg alloys are not only the main factors that determine the strength and corrosion performance [3,4,5], but also strongly affect the fatigue performance [6,7]. Therefore, it is necessary to investigate the precipitate characteristics and microscopic mechanism of 7020 aluminum alloy with optimal fatigue performance.

The types, sizes and distribution characteristics of the precipitates of Al-Zn-Mg aluminum alloy can be adjusted through different aging processes. The precipitates in the alloy mainly include GP zone, η’ metastable phase and η (MgZn_2_) equilibrium phase [8]. 

There are two types of Guinier Preston (GP) zone, namely GPI zone and Guinier Preston II (GPII) zone. The GP zone, which is the segregation region of Zn and Mg atoms, can be obtained at low temperature or short time high temperature aging [9]. The thickness of the GP zone is about 1–3 nm and it is completely coherent with the aluminum matrix [10]. The main precipitate of the alloy during aging is η’ at 120–170 °C, and is η phase at 170–250 °C [11]. The η’ is mainly transformed from the GP zone, and can also directly nucleate and precipitate from the supersaturated solid solution [12]. The η phase is an equilibrium phase and is non-coherent with the aluminum matrix. Aging characteristics also include GBP and PFZ. With the deepening of the aging, the GBP is larger and more intermittent, and the PFZ is wider [13,14]. The precipitates in under-aged alloys are mainly GP zone. The peak-aged alloy has GP zones and fine η’ phases within the grains and have continuous precipitates on grain boundary. When the alloy is over-aged, the η’ phases grow up and the η phases appear [15]. At present, the main methods for obtaining different precipitate characteristics of aluminum alloys are single-stage aging, two-stage aging, retrogression and re-aging, and secondary aging, etc. [16,17]. T73, T74 and T76 belong to two-stage over-aging and the degree of over-aging decreases in turn. The first step of the two-stage aging is carried out at a lower temperature. The purpose is to provide nucleation core for subsequent aging, and to reduce the number of direct precipitates of the equilibrium phase. The second step aging at a relatively high temperature can convert the GP zone to η’ phase, promote coarsening and intermittent GBPs, and widen the PFZ to improve the corrosion resistance [15]. The performance change of the two-stage aging control material is relatively wide, and it has been widely used in the heat treatment of aluminum alloys.

Two-stage aging can obtain different sizes of precipitates in under-aged, peak-aged and over-aged alloys. GP zones and fine η’ phases in the under-aged alloy can be sheared by dislocations during fatigue cyclic loading [18]. When fatigue microcracks propagate, the dislocations generated by fatigue cycles mainly slide along a favored slip plane in the grains. Most of the precipitates in the under-aged alloy can be repeatedly sheared by the fatigue crack tip dislocations. This mechanism makes the fatigue crack path highly serrated and tortuous [19,20]. Reasons such as reversible dislocation motion, the plastically induced crack closure (PICC) and RICC make fatigue crack propagation for under-aged alloy very slow [21]. Chen et al. [22] revealed wide and soft PFZ acted as a preferred route for intergranular crack propagation, and the soft PFZ degraded fatigue crack propagation resistance. In addition, GBPs promote fatigue crack nucleation at their interface and can accelerate crack propagation. Suresh et al. [23,24,25] have shown that as the degree of aging deepens, the larger size η phases generating during aging are easy to oxidize and improve the closing effect of particle clamps at the crack tip, which reduce the FCGR. Desmukh et al. [18] found that the 7010 aluminum alloy has the slowest FCGR for near-peak aged alloy and the fastest FCGR for over-aged alloy. But they also pointed out that coarse and unsheared η′ phase and η phase promoted uniform plastic deformation in the over-aged alloy, which can also improve the fatigue strength. These studies indicate that both extremely small precipitates and larger precipitates can improve alloy fatigue properties, although other properties such as strength and corrosion properties may be different. Therefore, it is necessary to study the mechanism of precipitates relative fatigue performance in order to find the most optimized process to manufacture materials with good fatigue crack propagation resistance and high fatigue strength. The effect of microstructure on fatigue performance is discussed in this paper for five artificially aged 7020 aluminum alloys with distinct precipitate characteristics.

## 2. Experimental Procedures

### 2.1. Material and Treatment

The composition of the materials used in the present study is listed in Table 1. The alloy was supplied in extruded online quenching condition. The aging processes shown in Table 2 were conducted, in order to investigate the influence of precipitate distribution characteristics on the fatigue performance of the alloy. The extruded profiles were parked 72 h at room temperature before artificial aging.

### 2.2. Microstructural Characterization

Optical microscopy by OLYMPUS BX51M (Olympus, Tokyo, Japan) was used to characterize grain structures. Surfaces of polished specimens were subsequently etched in Graff Sargent solution. Fatigue fracture surfaces and fatigue crack growth path of the specimens were analyzed by ZEISS EVO MA10 (ZEISS, Oberkochen, Germany) scanning electron microscope (SEM) with an operating voltage of 20 kV. Transmission electron microscope (TEM), along with selected area electron diffraction (SAED), was employed to characterize the aging microstructure and dislocations. Discs of Φ3 mm were punched from ~100 μm thin foils and then use twin-jet electro-polishing in a 20% nitric acid and 80% methanol solution at −20 °C to −30 °C to prepare for TEM observations. These thin discs were examined on FEI Tecnai G^2^ F20 TEM (FEI, Hillsboro, OR, USA) machine operating at 200 kV. Fatigue crack growth path of the fatigued specimens were examined by electron backscatter diffraction (EBSD) method. Specimens for EBSD analysis were electropolished in a solution consisting of 10% perchloric acid and 90% ethanol at 20 V. Then EBSD measurements were performed on the ZEISS EVO MA10 with an accelerating voltage of 20 kV.

### 2.3. Property Tests

Tensile tests were carried out according to ISO 6892 standard on three specimens for each aged alloy with a 40 mm gauge length. Tensile direction is parallel to the extruding direction. Tensile tests were performed at room temperature on an MTS Landmark (MTS, Faribault, MN, USA) type testing machine with a cross-head speed of 2 mm/min. The specimens for fatigue S-N curve tests were machined in conformity with ISO 1099 standard and then performed on SDS100 (SINOTEST, Changchun, China) electric servo -hydraulic fatigue testing machine. Five stress levels were designed for each aged alloy, and three specimens were tested at each stress level. All S-N curve tests were conducted at a stress ratio *R* (*K*_min_*/K*_max_) of −1 with a loading frequency of 60 Hz on the machine in air and at room temperature. Sampling along the extrusion direction and the size (in mm) of the middle section of the specimens is 20 × 12 × 10 (*L × W × B*). The geometry of the fatigue specimen with rectangular section is shown in Figure 1. Fatigue crack growth tests were carried out according to ASTM E647 standard. Compact tension (CT) specimens were prepared from the profiles with a size (in mm) of 62.5 × 60 × 10 (*L × W × B*). The CT specimen geometry for FCGR studies is shown in Figure 2. Three specimens were tested for each aged alloy. Fatigue crack growth tests were conducted at a stress ratio (*R = K*_min_*/K*_max_) of 0.1 with a loading frequency of 10 Hz on the MTS Landmark fatigue tester at room temperature in air environment. Fatigue crack propagation direction of the specimen is perpendicular to the extrusion direction.

## 3. Results

### 3.1. Grain Structure

Grain structure of the 7020 aluminum alloy profile is shown in Figure 3. Long fiber grains and small equiaxed grains can be observed in the alloy. The grain sizes (mean line intercepts) in the ED and ND directions of this alloy longitudinal section are about 31.8 ± 2.8 μm and 14.7 ± 1.2 μm, respectively.

### 3.2. Aging Characteristics

Representative bright-field TEM images and corresponding SAED patterns were present in Figure 4 for the under-aged and peak-aged alloys. Figure 4a shows a large amount of fine precipitates with a diameter of 1–2 nm distributed in the under-aged alloy. The SAED pattern taken from <001>_Al_ zone axis is also represented in Figure 4a. The diffraction spots at {1, (2n + 1)/4, 0}_Al_ indicate that the GPI zones exist in the under-aged alloy [26]. Figure 4b shows that small and continuous precipitates are distributed on the grain boundaries. The average size of the GBPs along the grain boundary is about 13.5 nm. No obvious PFZ was observed near the grain boundaries. TEM images and corresponding SAED pattern of the peak-aged alloy were shown in Figure 4c,d. The size of the precipitates in the peak-aged alloy is significantly larger than that of the under-aged state, with an average size of about 7.6 nm. The diffraction spots at 1/3 and 2/3 of {220}_Al_ were observed in <110> _Al_ projection. It indicates that the main precipitate of the peak-aged alloy is the η’ phase [26]. As shown in Figure 4d, there is obvious PFZ near the peak-aged grain boundary, with an average width of about 65.5 nm. The GBPs are relatively continuous along the grain boundary direction, and the average distance is increased compared to the under-aged alloy. The high resolution transmission electron microscope (HRTEM) images and corresponding SAED patterns for peak-aged alloy reveal that some fine GPI zones also appear in the alloy except η’ phase, as shown in Figure 5a,b. These GPI zones which are coherent with the Al matrix lattice, and η’ phases together ensure the peak-aged alloy has higher strength.

As shown in Figure 6a–f, with the increase of the degree of over-aging, the size of the precipitates in the alloy increases continuously. From T576 to T573, the size of intragranular precipitates increases from 8.5 nm to 10.7 nm, the width of PFZs also increases from 80.1 nm to 108.0 nm, and the GBPs interval become larger and larger. SAED pattern proves that η equilibrium phases appear in T573 alloy, as shown in Figure 6e. Combined with the TEM images, there is also a long strip of η phases precipitated in T574 condition. The precipitation characteristics analyses of different aged alloys are listed in Table 3. The data in the table were obtained by multiple representative TEM images.

### 3.3. Tensile Properties

Tensile properties for the different aged alloys at room temperature are presented in Figure 7. The under-aging alloy has the lowest yield strength of 214 MPa and the highest elongation of 15.6%. The yield strength and elongation of peak-aged alloy are 311 MPa and 13.6%, respectively. The yield strength of the under-aged alloy is 31.2% lower than the peak-aged, and the elongation is 14.7% higher. With the aging from T576 to T573, the yield strength of the alloy continues to decrease, and the elongation continues to increase.

### 3.4. Fatigue Strength

Figure 8 represents the fatigue S-N curves and fatigue strength of different aged alloys. Fitting fatigue cycles data of the different aged alloys according to Equation (1) (*A*, *B* and *E* are undetermined coefficients) [27], the following Equations (2)–(6) can be obtained:(1)$$\mathrm{lg}N=A+B\mathrm{lg}\left(S-E\right)$$
(2)$$\mathrm{Under}-\mathrm{aged}:\;\mathrm{lg}N=8.10-1.88\times \mathrm{lg}\left(S_{\max}-126.70\right)$$
(3)$$\mathrm{Peak}-\mathrm{aged}:\mathrm{lg}N=8.92-2.06\times \mathrm{lg}\left(S_{\max}-104.44\right)$$
(4)$$T576:\mathrm{lg}N=10.75-2.93\times \mathrm{lg}\left(S_{\max}-96.21\right)$$
(5)$$T574:\mathrm{lg}N=11.40-3.25\times \mathrm{lg}\left(S_{\max}-99.55\right)$$
(6)$$T573:\mathrm{lg}N=12.26-3.71\times \mathrm{lg}\left(S_{\max}-84.00\right)$$

When the material fatigue limit life *N* is taken as 10^7^ cycles, the fatigue strength of the different aged alloys can be calculated according to the fitting Equations (2) to (6), which are 130.6 MPa, 113.1 MPa, 115.3 MPa, 122.1 MPa and 110.2 MPa respectively, as shown in Figure 8b. The under-aged alloy has the highest fatigue strength, and it is 15.5% higher than that of the peak-aged alloy. As the degree of over-aging deepens, the fatigue strength increases firstly and then decreases.

### 3.5. Fatigue Crack Growth Behavior

Figure 9 displays the comparison of FCGR curves of the different aged alloys based on the applied stress intensity ranges. The FCGR of under-aged alloy is significantly lower than the peak-aged alloy in the stages Ⅰ and Ⅱ of fatigue crack propagation, as shown in Figure 9a. At the low Δ*K*, the FCGR of the under-aged alloy is very slow and continuously decreasing. As the Δ*K* increases, the differences of FCGR for the under-aged and peak-aged alloy narrow down. The specimen fractures at about 26 MPa·m^1/2^ of Δ*K*, which is much earlier than that of the peak-aged alloy, because of the lower yield strength of the under-aged alloy. Among the over-aged alloys, the FCGR of the moderately over-aged T574 alloy is the slowest, T576 alloy is the second, T573 alloy is the fastest, as shown in Figure 9b. The linear portion of *da*/*dN* versus Δ*K* in Figure 9, i.e., the Paris region is found to obey the power law relationship $da/dN=C{\left(\Delta K\right)}^{n}$ [28], where *C* and *N* are the Paris constants. The Paris constants and the FCGR of the different aged alloys have been calculated and presented in Table 4. The *da*/*dN* values at specified Δ*K* levels between 10 and 20 MPa·m^1/2^ were determined to make the comparisons. The FCGR of the under-aged and moderately over-aged alloys is 58.1% and 35.6% slower than that of the peak-aged alloy at Δ*K* of 10 MPa·m^1/2^, respectively. As the degree of over-aging deepens, from T576 to T574 to 73, the FCGR decreases firstly and then increases.

The fatigue fracture surfaces in near-threshold (Δ*K* of about 7 MPa·m^1/2^) and Paris regime (Δ*K* of about 15 MPa·m^1/2^) of the under-aged alloy were exhibited in Figure 10. Clear faceting can be observed at low Δ*K* for the alloy in Figure 10b. The gradual transition between ductile and crystallographic crack growth has been observed at Paris regime. Significant secondary cracks appeared in the under-aged alloy in the near-threshold regime, as arrowed in Figure 10a. The formation of the secondary cracks lead to lower FCGR because of reducing the fatigue driving force [29]. The size of shear facets divided by the tearing ridges in each stage of fatigue crack growth is very small. However, the shear facets in the Paris regime are larger than those in the near-threshold regime. Gupta [30] revealed that plastic strain accumulation during cyclic loading is responsible for the different sizes of the facets. This can explain the above observations well. The distance between adjacent fatigue striations corresponds to the average crack growth rate per cycle for cyclic loads [31]. The fatigue striations of under-aged alloy are very narrow and flat, indicating that the FCGR is very slow, as shown in Figure 10d.

Figure 11 shows the representative fatigue fracture surfaces of the T574 and T573 alloys in Paris regime. The fatigue facets of these two alloys are larger and rounder than that of under-aged alloys, and the facets of T574 alloy is relatively smaller. The tearing ridges of T573 alloy is smoother than that of T574 alloy. Comparing the fatigue striations of the two alloys, it can be seen that in the Paris regime, the FCGR of T573 alloy is faster than that of T574, which is coincident with the test result in Figure 9b.

## 4. Discussion

### 4.1. Fatigue Crack Initiation

The main microstructure factors that affect the fatigue performance of aluminum alloys are grain size, intermetallic particles, grain orientation and precipitates [32,33,34]. Various aging heat treatment had little influence on the size and orientation of grain [35]. Dislocation-precipitate interactions were significant in the fatigue crack tip region, so this research highlights the effect of precipitate while ignoring other factors effect. There are mechanisms for cutting and bypassing when dislocations pass through the precipitates. Kovacs [36] performed theoretical calculations on the shear stress when dislocation cutting and bypassing the precipitates. The calculation result is that the strengthening effect when the dislocation cuts the precipitate increases with the increase of particle size and volume fraction. However, the strengthening effect when bypassing mechanism decreases as the size of the precipitate increases and the volume fraction decreases. Therefore, there is a critical size. When the size of the precipitate is smaller than the critical size, the dislocation shears the precipitate, and when it is larger than the critical size, the dislocation bypasses the precipitate. For the precipitates in Al-Zn-Mg alloys, this critical radius is about 3 nm [16,36]. The size of the GPI zones in the under-aged alloy is between 1 and 2 nm (as listed in Table 3), so dislocations can shear it and glide reversibly during cyclic loading. This phenomenon also occurs in the under-aged Al-Cu-Mg alloy [37]. During cyclic loading, dislocations are not easy to accumulate and cause stress concentration. Therefore, the time of fatigue crack initiation is delayed at low stress, improving the fatigue strength of the under-aged alloy, as shown in Figure 8. However, the fatigue life under high stress of the under-aged alloy is shortened due to its lowest yield strength. Because the size (above 7.6 nm) of the precipitate in the peak-aged and over-aged alloys is larger than the critical size, dislocation can only bypass the precipitate. These occurrences were also observed in Figure 12. It can be seen from the figure that the dislocations in the under-aged alloy do not accumulate in the vicinity of the grain boundary after 10^7^ cycles loading. The dislocations pass through the precipitates in a straight line. After 10^7^ cycles loading, a large amount of dislocations in the peak-aged alloy has accumulated near the grain boundary.

Coarser η’ particles in the peak-aged and over-aged alloy were a kind of semi-coherent unshearable precipitate in nature [38]. As a result, irreversible dislocation motion is easy to cause dislocation aggregation, causing stress concentration and accelerating crack nucleation. The schematic diagram of the interaction process between dislocation and precipitation characteristics during fatigue crack initiation of peak-aged and over-aged alloy is illustrated in Figure 13. In the over-aged alloy, the distance of GBPs is greater than the peak-aged alloy. The dislocation generated by fatigue stress easily crosses the grain boundary. The concentration effect of dislocations and the degree of stress concentration is reduced, which is beneficial to slow the fatigue crack initiation [19]. When the dislocation passes through the PFZ, the wider the PFZ is, the more the stress get relaxed, and the more difficult for the fatigue cracks to initiate. In addition, strength difference between the grain interior and PFZ near the grain boundaries of over-aged alloy becomes smaller. This in turn results in a higher deformation resistance of the grain boundary [18]. A combination of these factors makes fatigue crack of over-aged alloy more difficult to initiate, and improves the fatigue strength as compared to the peak-aged alloy, as shown in Figure 8. However, as the yield strength of the T573 alloy was too low, leading the ability to resist plastic deformation weakened, the fatigue strength was also reduced.

### 4.2. Fatigue Crack Growth Mechanism

Figure 14 shows the fatigue crack growth path at the Δ*K* of about 9 MPa·m^1/2^ of the under-aged alloy. Due to the extremely small shearable GPI zones in the under-aged alloy, dislocations shear them to localize in planar bands of varying coarseness and glide reversibly during cyclic loading. This planar-reversible slip is found to promote fatigue crack propagation resistance by increased crack tip deflection, crack closure and reduced damage accumulation [39,40]. This mechanism makes the crack propagation path in the under-aged alloy very tortuous and have many secondary cracks, as arrowed in the figure. On the one hand, when the fatigue crack path is tortuous, it can improve RICC effect and increase the total length of the crack, causing longer time to extend from crack initiation to fracture. On the other hand, when the crack propagation direction deviates from the vertical direction of the applied stress, the effective driving stress intensity factor Δ*K*_eff_ of the crack tip will be less than the corresponding theoretical straight crack [41,42], so the propagation speed of the deflection crack becomes slower. There is also a small fatigue cracks phenomenon in under-aged alloy [43,44,45]. The stress field in the plastic zone of the crack tip generated by a small crack is within a range of grain size. Due to the sub-grain or grain boundary, the crack is deflected or bifurcated, and the driving force of the crack tip propagation changes, resulting in fatigue crack growth deceleration. And the PICC effect of the under-aged alloy is also more significant because of its highest plasticity (as shown in Figure 7). Based on the above analysis, under-aged alloys with extremely fine precipitates have the best fatigue crack propagation resistance (as shown in Figure 9a).

The fatigue crack growth path of the peak-aged and T573 alloy are shown in Figure 15 and Figure 16, respectively. The fatigue cracks in the peak-aged and over-aged alloy mainly propagate transgranular, and occasionally propagate intergranular at the small grains’ boundary. Fatigue crack deflections at 1 to 5 are due to the influence of grain boundaries, as shown in Figure 15b. It can be seen from Table 4 that the peak-aged and over-aged alloys have a crack propagation distance of 0.103 to 0.160μm per fatigue cyclic loading at Δ*K* of 10 MPa·m^1/2^. The previous analysis shows that the average grain size in the ND direction of the alloy longitudinal section is about 14.7μm. When fatigue crack propagates through the grains, it takes about 92 to 143 times to pass through a grain. In addition, the width of PFZs in different aging alloys is between 65.5 and 108.0 nm, which is very narrow compared to the grain size. Therefore, the dominant factor affecting the FCGR is the distribution characteristics of the intragranular precipitates.

The linear density and distance of the intragranular precipitates in the peak-aged and over-aged alloys (Figure 4 and Figure 6) are counted, as listed in Table 5. Combined with the FCGR data of each aged alloy in Table 4, the number of precipitates included in the per cyclic loading at Δ*K* of 10 MPa·m^1/2^ can be calculated. The calculation results are also shown in Table 5. Comprehensive the data in the table, a schematic diagram of crack propagation based on the distribution characteristics of the precipitates along the fatigue crack growth path is shown in Figure 17. The size of intragranular precipitates in the peak-aged alloy is small and the distance between them is narrow, indicating a high probability of dislocations interactions with precipitates [7]. Since the size of the precipitates in the peak-aged is larger than the critical size, dislocation can only bypass the precipitate, leaving dislocation loops behind the precipitates. As the cyclic loading, dislocations tend to pile up around the precipitates, causing stress concentration. And the narrow region of the soft matrix between the precipitates limits the reversibility of dislocation slip. In addition, the plasticity of the peak-aged alloy reduced significantly, which leads to a serious weakening of the PICC effect during crack growth. So, the FCGR is faster (as shown in Figure 9a). The larger precipitate size and distance within the grains of moderately over-aged alloy (T574) offered a relatively wider slipping distance of dislocations, thereby promoting the slip reversibility and relaxing stress of the crack tip, finally reducing the fatigue damage accumulation. And the larger size η phases are easy to oxidize inducing oxide closure and promote the RICC effect by particles clamping at the crack tip [25]. Therefore, the FCGR of moderately over-aged alloy is slower. When the low-strength region of the soft matrix is too wide, the resistance to plastic deformation becomes low, and the crack grows faster in it. Besides, a small part of fatigue crack propagated along the large deformed grain boundary in the depth over-aged (T573) alloy, as arrowed in Figure 16b. This is mainly because the coarse GBPs are easy to concentrate the stress and the excessively wide PFZs make the grain boundary yield strength low, so under these conditions, it is easy to cause intergranular cracks and induce fatigue crack propagation along the grain boundary [22,46]. This is one of the reasons why the FCGR of T573 alloy is fast.

## 5. Conclusions

The effect of different precipitate microstructures on fatigue fracture behavior of 7020 aluminum alloy was investigated in this study. The main conclusions could be drawn as follows:

1.The GPI zones below the critical size in the under-aged alloy can be repeatedly sheared by dislocations, making it difficult for fatigue cracks to initiate. Moreover, the fatigue crack growth path is tortuous when the precipitates can be repeatedly sheared by dislocations caused by cyclic loading, and the FCGR is 58.1% slower than that of the peak-aged alloy at Δ*K* of 10 MPa·m^1/2^.2.The alloys with unshearable precipitates, continuous GBPs and narrow PFZs are prone to initiate fatigue crack and reduce fatigue strength.3.Unshearable precipitates with moderate size in the over-aged alloy improve the RICC effect by particles clamping at the crack tip. The soft matrix with appropriate width between the precipitates can promote the slip reversibility and relax the crack tip stress. The fatigue strength of the moderately over-aged alloy is about 122.1 MPa at 10^7^ cycles of loading, and the FCGR is 35.6% slower than that of the peak-aged alloy at Δ*K* of 10 MPa·m^1/2^. Moderate over-aging improved fatigue performance of the alloy.

## Figures and Tables

**Figure 1 materials-13-03248-f001:**
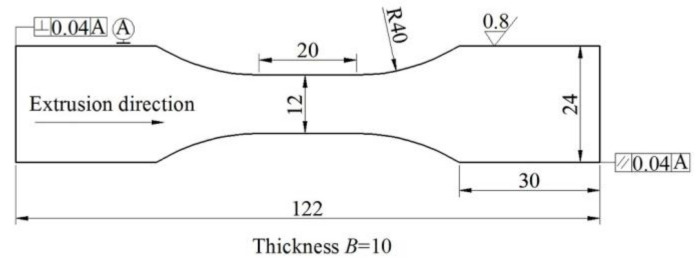
Fatigue strength specimen geometry (mm).

**Figure 2 materials-13-03248-f002:**
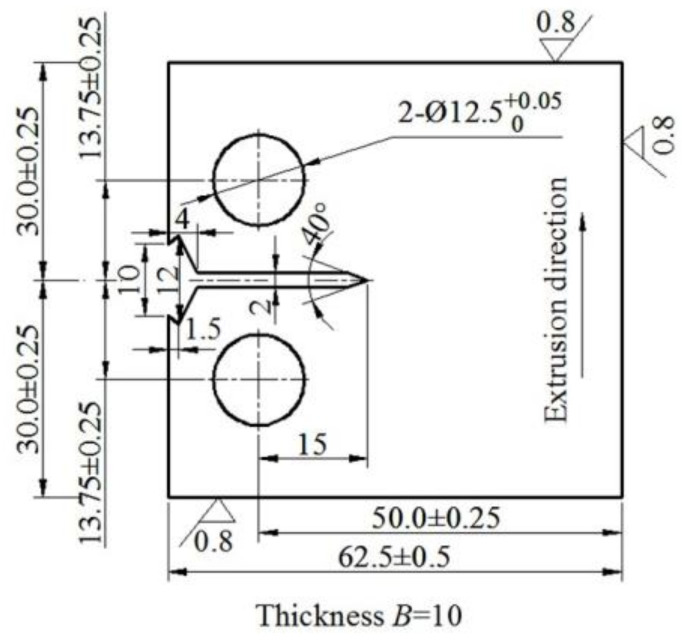
Compact tension specimen geometry for the FCGR studies (mm).

**Figure 3 materials-13-03248-f003:**
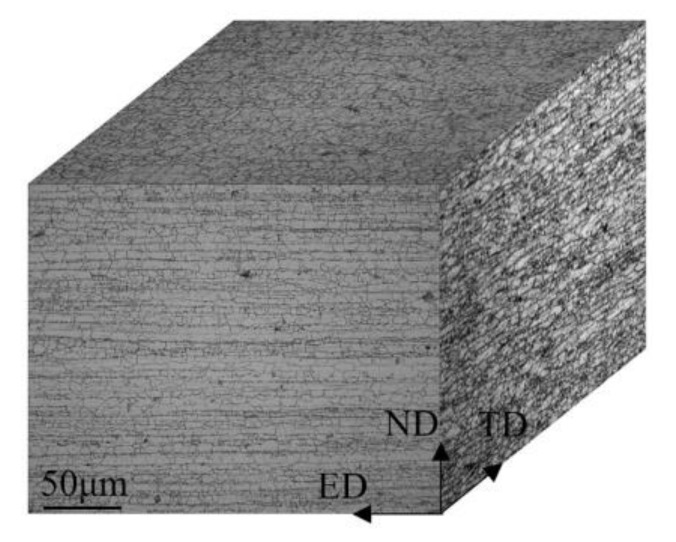
Optical micrograph of 7020 aluminum alloy.

**Figure 4 materials-13-03248-f004:**
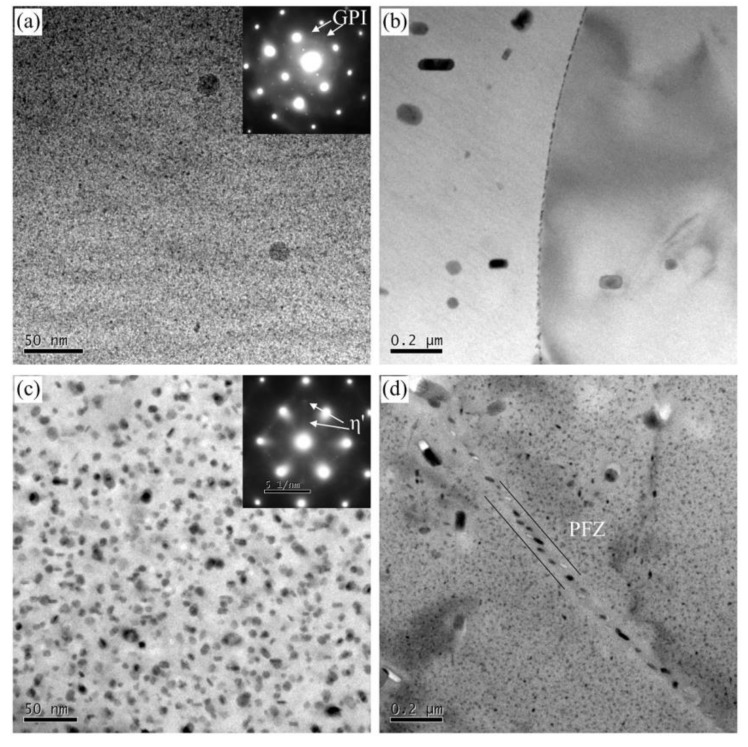
TEM bright-field images and corresponding SAED patterns for (**a**,**b**) under-aged alloy, (**c**,**d**) peak-aged alloy.

**Figure 5 materials-13-03248-f005:**
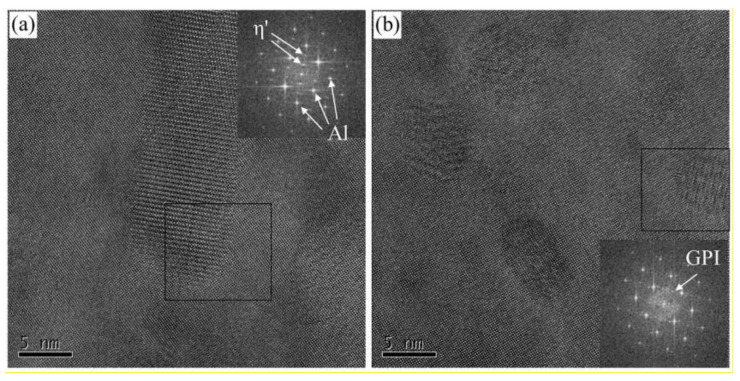
HRTEM images and corresponding SAED patterns for peak-aged alloy. (**a**) η’ phase; (**b**) GPI zone.

**Figure 6 materials-13-03248-f006:**
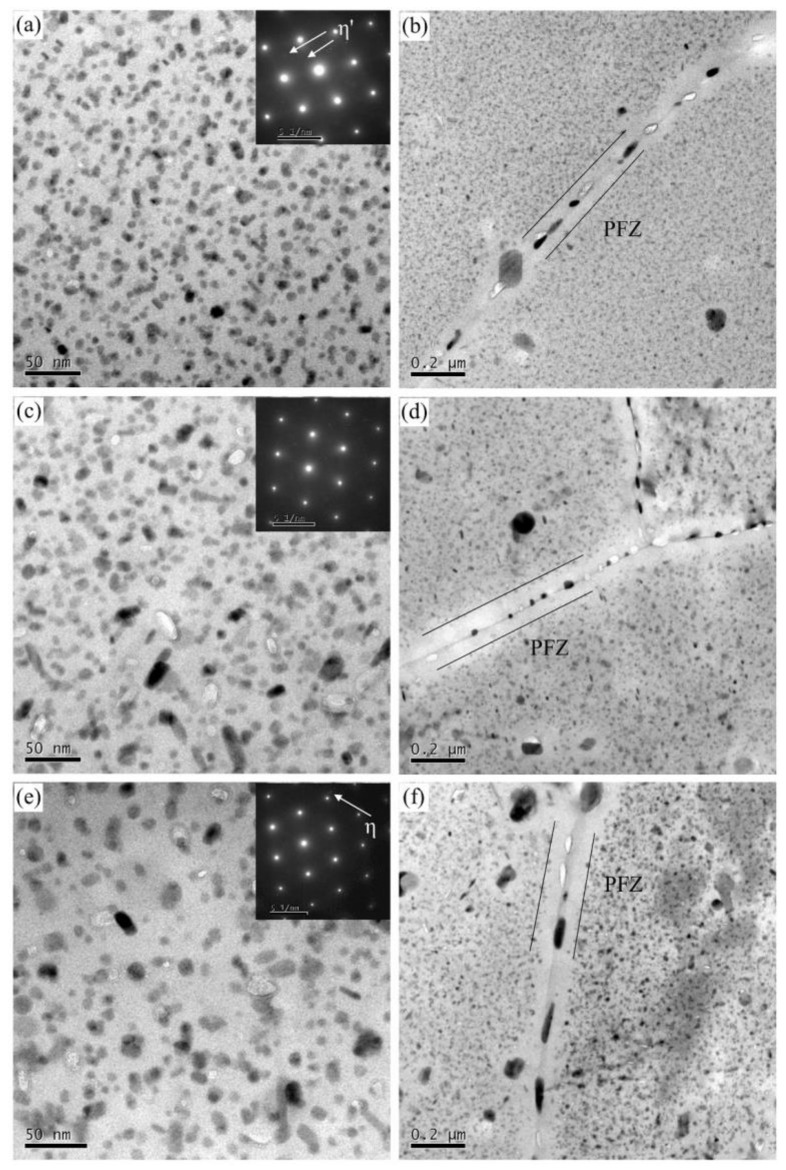
TEM bright-field images and corresponding SAED patterns for (**a**,**b**) T576 alloy, (**c**,**d**) T574 alloy and (**e**,**f**) T573 alloy.

**Figure 7 materials-13-03248-f007:**
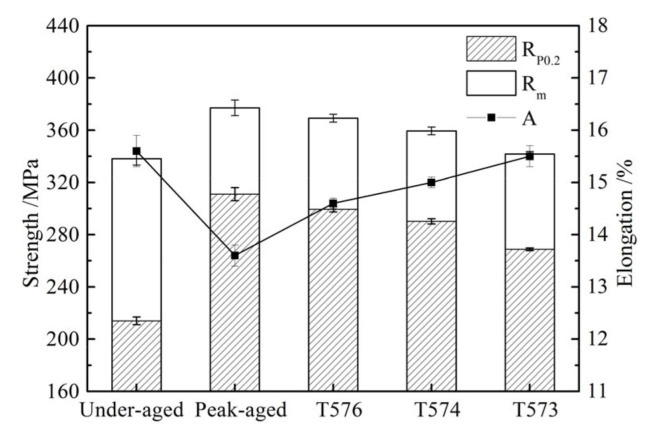
Tensile properties for the different aged alloys at room temperature.

**Figure 8 materials-13-03248-f008:**
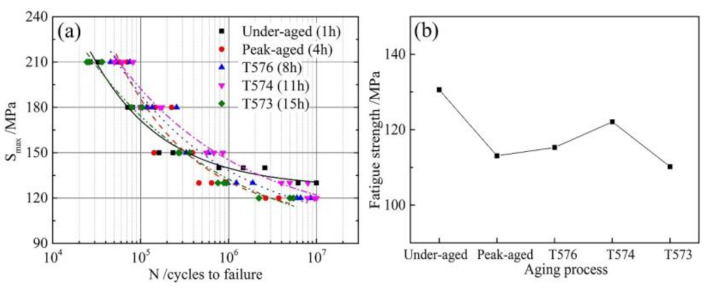
(**a**) S-N curves and (**b**) Fatigue strength at 10^7^ cycles of different aged alloys.

**Figure 9 materials-13-03248-f009:**
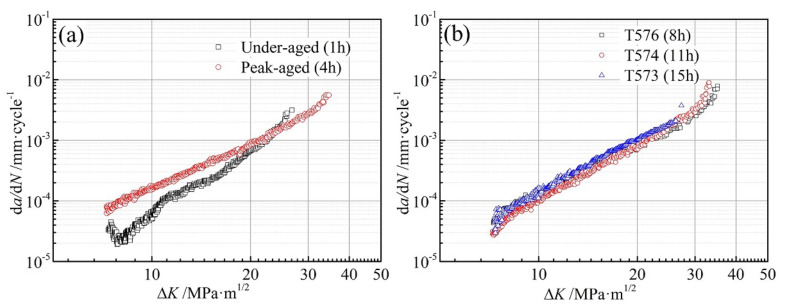
FCGR curves of (**a**) under-aged and peak aged and (**b**) T576, T574 and T573 alloys.

**Figure 10 materials-13-03248-f010:**
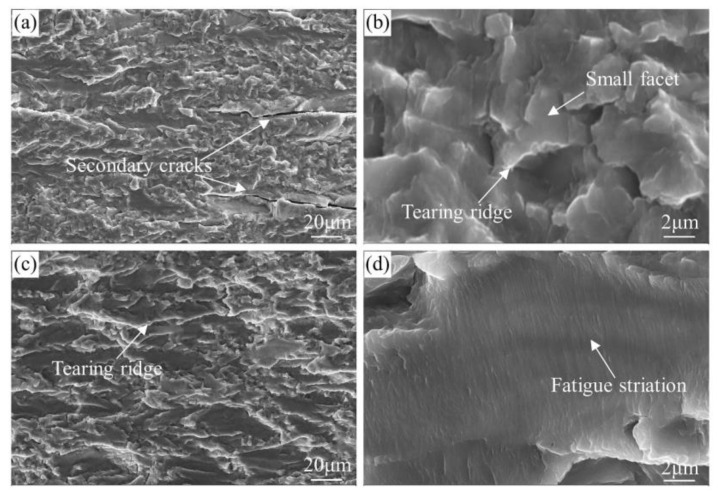
SEM fractographs characterizing the fatigue fracture surfaces of the under-aged alloy at Δ*K* of about (**a**,**b**) 7 MPa·m^1/2^ and (**c**,**d**) 15 MPa·m^1/2^.

**Figure 11 materials-13-03248-f011:**
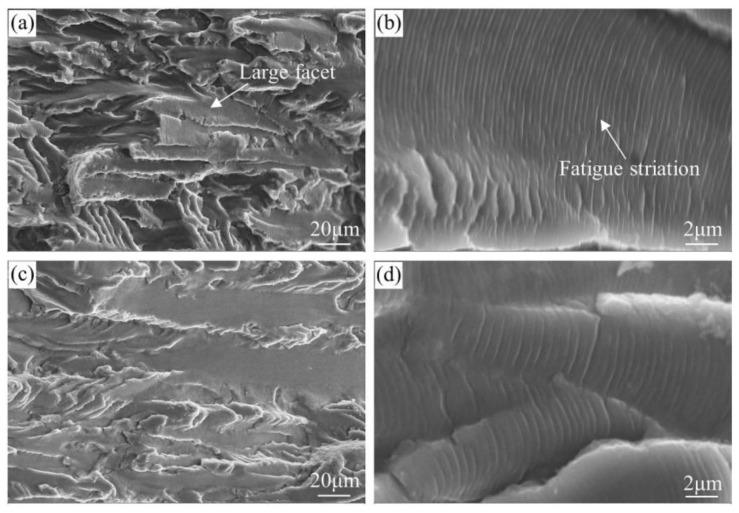
SEM fractographs characterizing the fatigue fracture surfaces of the (**a**,**b**) T574 and (**c**,**d**) T573 alloy at Δ*K* of about 15 MPa·m^1/2^.

**Figure 12 materials-13-03248-f012:**
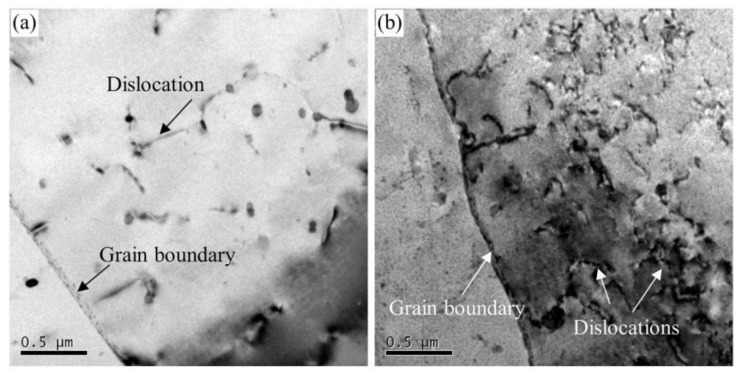
Dislocations micrographs of (**a**) under-aged alloy and (**b**) peak-aged alloy after 10^7^ cycles loading.

**Figure 13 materials-13-03248-f013:**
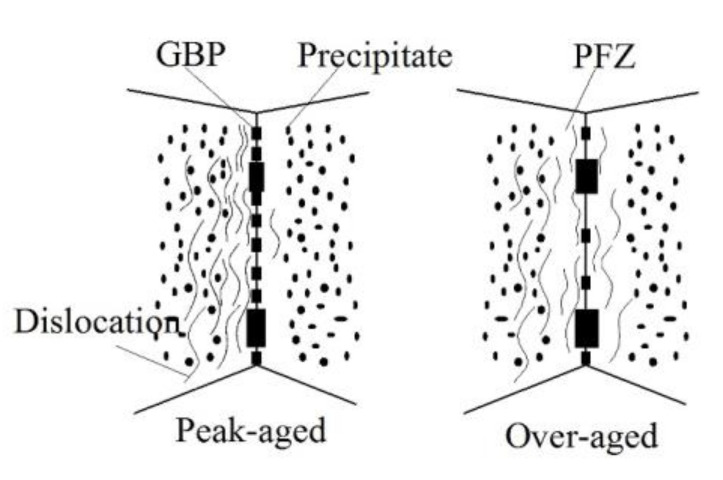
Schematic representation of dislocation motion process during fatigue crack initiation.

**Figure 14 materials-13-03248-f014:**
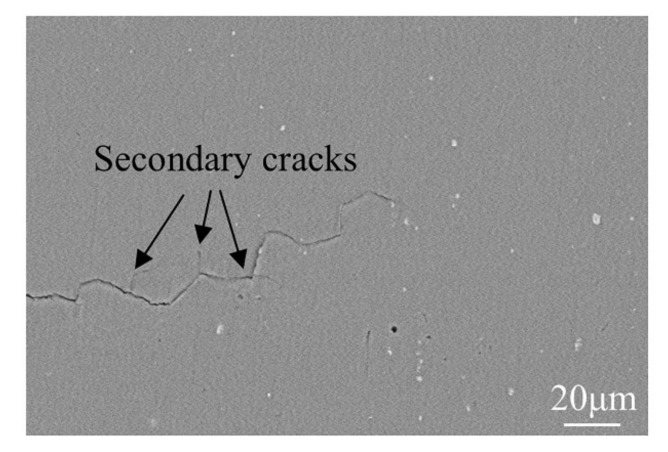
Fatigue crack propagation path at the Δ*K* of about 9 MPa·m^1/2^ of the under-aged alloy.

**Figure 15 materials-13-03248-f015:**
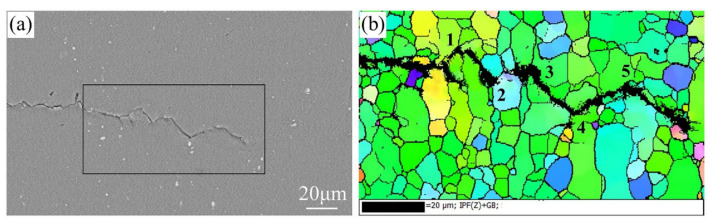
Fatigue crack propagation path at Paris regime of the peak-aged alloy (**a**) SEM and (**b**) EBSD.

**Figure 16 materials-13-03248-f016:**
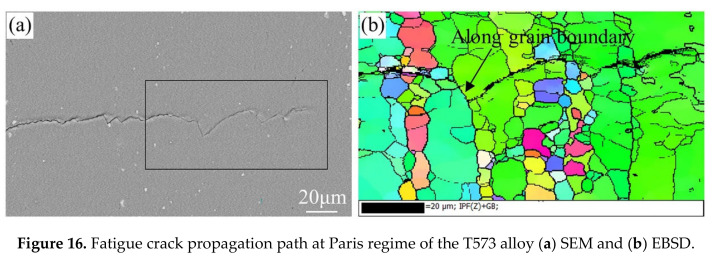
Fatigue crack propagation path at Paris regime of the T573 alloy (**a**) SEM and (**b**) EBSD.

**Figure 17 materials-13-03248-f017:**
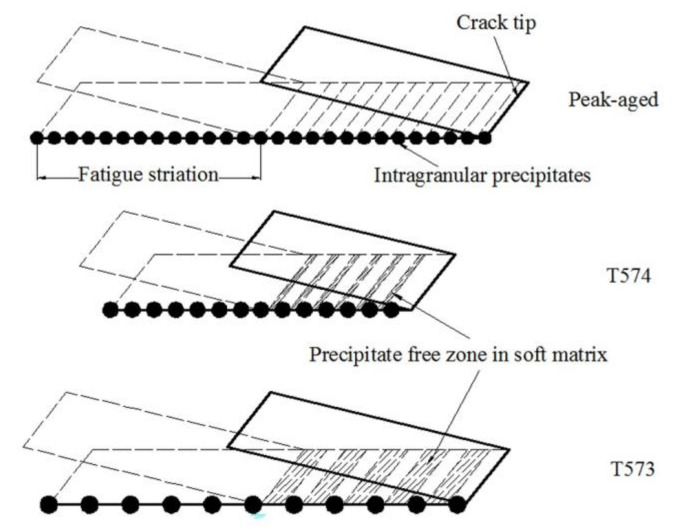
Schematic representation of precipitation characteristics affecting fatigue crack growth in peak-aged and over-aged alloys.

**Table 1 materials-13-03248-t001:** Chemical composition of the 7020-aluminum alloy (wt.%).

Zn	Mg	Cu	Mn	Cr	Ti	Zr	Fe	Si	Al
4.43	1.16	0.10	0.32	0.21	0.05	0.14	0.18	0.09	Bal.

**Table 2 materials-13-03248-t002:** Heat treatment schedule.

Alloys	Heat Treatment
Under-aged (1 h)	90 °C/12 h + 170 °C/1 h
Peak-aged (4 h)	90 °C/12 h + 170 °C/4 h
T576 (8 h)	90 °C/12 h + 170 °C/8 h
T574 (11 h)	90 °C/12 h + 170 °C/11 h
T573 (15 h)	90 °C/12 h + 170 °C/15 h

**Table 3 materials-13-03248-t003:** Precipitation characteristics analyses of different aged alloys.

Alloys	Intragranular Precipitate (nm)	GBP (nm) (Along Grain Boundary)	PFZ (nm)	Main Types
Under-aged	1–2	13.5 ± 1.5	-	GPI
Peak-aged	7.6 ± 0.2	26.8 ± 2.7	65.5 ± 1.7	η’, GPI
T576	8.5 ± 0.3	31.0 ± 3.3	80.1 ± 2.1	η’
T574	9.5 ± 0.5	37.4 ± 3.4	96.8 ± 2.6	η’, η
T573	10.7 ± 0.7	50.6 ± 4.8	108.0 ± 3.5	η’, η

**Table 4 materials-13-03248-t004:** Fatigue crack growth data of the different aged alloys.

Alloys	*C*	*n*	*da*/*dN* = *C*(Δ*K*)*^n^* (mm/Cycle)		
			Δ*K* = 10 MPa·m^1/2^	Δ*K* = 15 MPa·m^1/2^	Δ*K* = 20 MPa·m^1/2^
Under-aged	3.14 × 10^−8^	3.33	6.71 × 10^−5^	2.59 × 10^−4^	6.75 × 10^−4^
Peak-aged	5.19 × 10^−7^	2.49	1.60 × 10^−4^	4.40 × 10^−4^	9.01 × 10^−4^
T576	2.11 × 10^−7^	2.78	1.27 × 10^−4^	3.92 × 10^−4^	8.73 × 10^−4^
T574	8.98 × 10^−8^	3.06	1.03 × 10^−4^	3.57 × 10^−4^	8.60 × 10^−4^
T573	1.86 × 10^−7^	2.88	1.41 × 10^−4^	4.54 × 10^−4^	1.04 × 10^−3^

**Table 5 materials-13-03248-t005:** Analyses of precipitates contained in per fatigue cycle of peak-aged and over-aged alloys.

Alloys	Linear Density ^1^/μm	Distance ^2^/nm	FCGR mm/Cycle	Precipitates Number/Cycle
Peak-aged	85.0	4.2 ± 0.1	1.60 × 10^−4^	13.6
T576	76.7	4.6 ± 0.1	1.27 × 10−4	9.7
T574	67.3	5.4 ± 0.3	1.03 × 10^−4^	6.9
T573	45.0	11.8 ± 0.5	1.41 × 10^−4^	5.8

^1^ Number of precipitates on the unit line. ^2^ Average matrix width between two precipitates.

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
