# Peer review of "Mechanism of Precipitate Microstructure Affecting Fatigue Behavior of 7020 Aluminum Alloy"

_materials, 2020, doi:10.3390/ma13153248_

Round 1
Reviewer 1 Report
The manuscript "Mechanism of precipitate microstructure affecting fatigue behavior of 7020 aluminum alloy" studies the fatigue crack behavior in 7020 aluminum alloy in different conditions. Particularly, underaged, peak-aged and three over-aged conditions were chosen. Authors study the microstructure of grain boundaries area and establish the relationship between treatment conditions, microstructure and cracking behavior of these alloys.
The manuscript is written well and solid, substantial background of the problem is presented.
However, reviewer has few questions and remarks.
1. Table 1 - The contents of Cu, Ti, Zr, Si are out of range for 7020 alloy. Does it make any difference in this study?
2. P3 L106 - What does "parallel specimens" mean?
3. P4 paragraph 3.2 - "GPI" should probably be "GP";
Reviewer would recommend avoid using word "obviously", especially this often;
4. P4 L133 - What is the error for 13.5 nm value?
5. P5 Fig. 2 - SAED patterns are small, the reflects from n-phase cannot be seen.
6. P6 L152 - What is the error for the precipitate size (8.5nm and 10.7nm)?
7. Table 3 - Table does not contai any error values.
8. P10 L254 - The sentence "The dislocations ..." is not clear.
9. P13 L325 - The sentence "As the cyclic loading ..." is not clear.
10. P13 L334 - The sentence "But in the depth ..." is not clear.
11. P12 L326 - "narrow soft matrix", probably "narrow region of the soft matrix". Same for the P13 L336.
12. P13 L336 - "resistance ... weak". Resistance should probably be low of high, not weak.
13. Table 5 - The content of the table requires explanation. What is the value of "Distance" and "Linear density"? Distance value also doesn't have errors. Should it be correct that the error is lower than difference between 4.2 and 4.6 nm?
14. Fig. 15 - from the figure it is not clear what is the difference. Figure requires to be redone, the description needs to be more detailed.
Reviewer 2 Report
Presented work is valuable, but have minor shortcomings. Presented relation between artificial aging to the fatigue life or the fatigue crack growth is purposeful. It is interesting, is it possible to get a quantitative relationship between the size of the precipitate and fatigue strength or FCGR? Other comments are the following:
In chapter 2.3 Figures of the specimens should be given.
Line 123 It should be given a standard deviation of the grain size.
What mean GPI zone? I haven't found it in the text.
In chapter 3.4 authors didn't give information about how many samples were used to determine the S-N curve for each material. The same is in chapter 3.5.
Why did the authors compare the fatigue strength for 10^7 cycles? In reference (18) the fatigue strength is compared for 10^6 cycles. The results can be different for other fatigue life. In eq. (1)-(6) there is parameter E, which can be defined as "fatigue limit". Can it be compared?
Figure 7, (a) and (b) should be named. Maybe, it will be better, if it will be one diagram?
